# A novel infrared imager for studies of hydroxyl and oxygen nightglow emissions in the mesopause above northern Scandinavia

Peter Dalin[1], Urban Brändström[1], Johan Kero[1], Peter Voelger[1], Takanori Nishiyama[2], Trond Trondsen[3], Devin Wyatt[3], Craig Unick[3], Vladimir Perminov[4], Nikolay Pertsev[4], Jonas Hedin[5]

[1]Swedish Institute of Space Physics (IRF), Kiruna, Sweden
[2]National Institute of Polar Research, Tokyo, Japan
 Keo Scientific Ltd., Calgary, Canada
[4]A.M. Obukhov Institute of Atmospheric Physics, Moscow, Russia
[5]Stockholm University, Department of Meteorology, Stockholm, Sweden

*Correspondence to*: Peter Dalin (pdalin@irf.se)

**Abstract.** The paper describes technical characteristics and presents first scientific results of a novel infrared imaging system (imager) for studies of nightglow emissions coming from hydroxyl (OH) and molecular oxygen ($O_2$) layers in the mesopause region (80-100 km) above northern Scandinavia. The OH imager was put into operation in November 2022 at the Swedish Institute of Space Physics located in Kiruna (67.86°N, 20.42°E, 400 m altitude). The OH imager records selected emission lines in the OH (3-1) band near 1500 nm to obtain intensity and temperature maps at around 87 km altitude. Also, the OH imager registers infrared emissions coming from the $O_2$ IR A-band airglow at 1268.7 nm in order to obtain $O_2$ intensity maps at a slightly higher altitude around 94 km. This technique allows tracing wave disturbances both in horizontal and vertical domains in the mesopause region. Validation and comparison of the OH (3-1) rotational temperature with collocated lidar and Aura/MLS satellite temperatures are performed. First scientific results obtained with the OH imager for the first winter season (2022-2023) are discussed.

## 1 Introduction

Besides a well-known scattering of sunlight on air molecules (Rayleigh scattering) and particle scattering, the Earth's atmosphere emits its own radiation 24 hours a day, year-round and on global scales. So-called airglow emissions occur in the mesosphere (50-90 km) and thermosphere (above 90 km). The most prominent airglow emissions are red and green lines of atomic oxygen as well as ultraviolet and infrared bands of molecular oxygen, hydroxyl (OH) emissions in the infrared part of the spectrum, yellow emission line of sodium (Na), ultraviolet and infrared bands of nitric oxide (NO) (e.g. Chamberlain, 1961; Khomich et al., 2008; Savigny, 2017).

Studies of airglow emissions provide important sources of information on physical and chemical states of the middle and upper atmosphere. In particular, temperature of emission layers in the middle and upper atmosphere can be determined by analyzing spectral bands and lines of airglow emissions. By analyzing spatial-temporal variations in the emission intensities and/or temperature one can investigate the short- and long-term dynamical state of the middle and upper atmosphere, which in turn is a hot topic of current atmospheric research (e.g., Fritts and Alexander, 2003; Plougonven and Zhang, 2014; Reid et al., 2014; Gavrilov et al., 2023; Wüst et al., 2023). The key role of the atmospheric

45 dynamical state belongs to waves, being a fundamental property of the atmosphere as they transport energy and momentum across great distances (Gossard and Hook, 1975).

  Observations of airglow emissions are used for inferring parameters of atmospheric waves since temperature and emission intensity fields in the mesosphere and thermosphere are directly modulated by gravity and planetary waves, solar thermal and lunar gravitational tides (e.g., Garcia et al., 1997;

50 Taylor et al., 1997; Pautet et al., 2005; Suzuki et al., 2008; Offermann et al., 2009; Gao et al., 2011; Perminov et al., 2018; Pertsev et al., 2021).

  Another very important aspect in atmospheric physics is a long-term monitoring of atmospheric parameters. The scientific community is wondering to what extent long-term changes in $H_2O$, $CO_2$, $N_2O$, $CH_4$, $O_3$ (greenhouse gases of anthropogenic and natural origin) are contributing to long-term

55 variability in the atmospheric temperature. While $CO_2$ is thought to endow to global warming on Earth, it also cools the middle atmosphere (Roble and Dickinson, 1989; Mlynzcak et al., 2022). The increased $H_2O$ concentration leads to warmer temperatures due to a strong positive water vapor feedback (Dessler et al., 2008). Methane oxidizes in the stratosphere and mesosphere producing an additional amount of water vapor in the middle atmosphere (Thomas and Olivero, 2001). The global monthly mean

60 atmospheric methane abundance demonstrates a constant increase since 1987 (https://gml.noaa.gov/ccgg/trends_ch4/), which may have resulted in the observed long-term trend in water vapor in the mesopause. Lübken et al. (2018) have obtained a summer mesopause cooling of ~1.2 K/decade for the period of 1960-2008 based on LIMA and MIMAS model simulations which included long-term evolutions of minor atmospheric species $CO_2$, $CH_4$, $H_2O$ and $O_3$. Thus, the long-term

65 temperature trend in the middle atmosphere is a complex function of several atmospheric components. Estimates of temperatures of the middle atmosphere are only possible with help of remote sensing instruments. That is why observations of airglow emissions are used for long-term monitoring of the temperature in the mesopause region (Semenov and Shefov, 1999; Espy and Stegman, 2002; Offermann et al., 2010; Ammosov et al., 2014; Kalicinsky et al., 2016; Perminov et al., 2018; Dalin et

70 al., 2020; French et al., 2020).

  More than fifty sites conducting spectroscopic and imaging airglow observations are presented at the Network for the Detection of Mesospheric Change (NDMC) which is a global program investigating climate change signals in the mesopause region (https://ndmc.dlr.de/). Li et al. (2018) have summarized a global distribution of all-sky airglow imager sites (see Table 1 in their paper and

75 references therein). Some of OH spectrographs and imaging instruments have a narrow field of view of 30 degrees and less such as the Ground-based Infrared P-branch Spectrometer instrument (GRIPS 6) in Oberpfaffenhofen, Germany (Schmidt et al., 2013), the Aerospace Nightglow Imager 2 (ANI2) in the Andes, Chile (Hecht et al., 2023), and the Spectral Airglow Temperature Imager (SATI) in Resolute Bay, Canada (Wiens et al., 1997). A number of OH imaging instruments measure OH emissions in

80 relative units without mesopause temperature derivations such as the ANI2, OH all-sky airglow imager in Kazan, Russia (Li et al., 2018), and the Fast Airglow IMager (FAIM) in Oberpfaffenhofen, Germany (Hannawald et al., 2016). Some of the OH imagers register OH emissions (including temperature derivations) without capturing infrared emissions coming from the $O_2$ layer such as the Advanced

Mesospheric Temperature Mapper (AMTM) at South Pole (Pautet et al., 2014), and the Near InfraRed Aurora Camera (NIRAC) in Svalbard, Norway (Nishiyama et al., 2024).

In the present paper, we describe a novel infrared wide-angle imaging instrument (in the following referred to as the OH imager) capable of registering emissions coming from two emitted layers (OH and $O_2$) and deriving the mesopause temperature. The OH imager is the first one of its kind installed in northern Scandinavia. The instrument measures infrared emissions of selected lines in the OH (3-1) band at wavelengths near 1.5 μm to produce intensity and temperature maps in the mesopause region at around 87 km altitude. Additionally, the OH imager records nightglow emissions from the $O_2$ IR A-band at 1268.7 nm to obtain $O_2$ emission intensity maps at a slightly higher altitude of about 94 km. Combining data from both layers allows tracing wave disturbances in 3-D space, i.e., horizontally and vertically in the mesopause region. The OH imager was installed in November 2022 at the Swedish Institute of Space Physics (IRF) in Kiruna (67.86°N, 20.42°E, 400 m altitude) on the eastern slope of the Scandinavian mountain range. Mountain gravity waves occur frequently at the location, thus the OH imager can be utilized to study the influence of such waves on the thermal and dynamical regime of the mesopause region over Kiruna. The OH registrations are part of the long-term research infrastructure commitment of the Kiruna Atmospheric and Geophysical Observatory (KAGO), part of the IRF, whose objective is to provide present and future scientists with long, continuous archived time series of data of highest possible scientific quality from its set of instruments.

In section 2, we describe technical characteristics of the OH imager as well as the principle of OH (3-1) rotational temperature measurements. In section 3, the temperature estimation and its uncertainty, calibration and validation technique are presented. Validation of OH (3-1) temperatures with the collocated Esrange lidar temperature measurements as well as a comparison of OH temperatures with simultaneous Aura/MLS satellite temperatures are shown in section 4. In section 5, first scientific results obtained with the OH imager are discussed. Finally, conclusions are given in section 6.

## 2 Instrument description

### 2.1 Theory and basic knowledge:

The hydroxyl (OH) airglow emission layer exists in the Earth's atmosphere between 75 and 100 km, with the highest concentration at about 87 km and the full width at half maximum of about 8-10 km (Baker and Stair, 1988; Savigny, 2017), although both of these parameters of the OH-layer change depending on geolocation and season (Perminov et al., 1999, 2018; Melo et al., 2000; Grygalashvyly et al., 2014). Its rotational-vibrational bands, covering the spectral region from 0.5 to 4.5 μm, result mainly from chemiluminescent processes (Bates and Nicolet, 1950):

$$O + O_2 + M \rightarrow O_3 + M$$

$$O_3 + H \rightarrow OH^* + O_2 + 3.3 \text{ eV}$$

where M is a molecule of nitrogen or oxygen, $OH^*$ is the OH vibrationally-excited molecule. Returning to their ground state, the $OH^*$ molecules emit visible and infrared radiation in the spectral range between 500 and 4500 nm.

The OH (8-3) band at 732-742 nm and the OH (6-2) band at 830-845 nm are commonly used in OH airglow observations, which are detectable by commercial CCD cameras. However, at high latitudes,

auroral emissions (molecular and atomic oxygen lines between 730 and 845 nm) have stronger intensities than those of the OH airglow bands (Christensen et al., 1978; Sivjee et al., 1979). Therefore, measurements of OH nightglow emissions in the Short Wave Infrared Range (SWIR) of the spectrum (900-1700 nm) are preferable in order to minimize auroral contaminations at high latitudes (Pautet et al., 2014; Nishiyama et al., 2021). For this purpose, the OH (3-1) band (1470-1550 nm) is used by the OH imager to measure nightglow emissions at high latitudes. It is important to note that the OH (3-1) intensity is by two orders of magnitude greater than intensities of the OH (8-3) and OH (6-2) bands. Also, the OH (3-1) emission is less affected by water vapor absorption. All these factors make a strong preference in choosing the OH (3-1) nightglow observation in Kiruna, being located underneath the auroral oval where auroral emissions occur frequently.

### 2.2 Instrument description

The main component of the OH imager is an infrared camera (Teledyne Princeton Instruments NIRvana 640) equipped with an Indium-Gallium-Arsenide (InGaAs) sensor for low-light scientific SWIR imaging applications. The NIRvana 640 camera employs a 640 x 512 InGaAs array, charged-coupled device (CCD), with a spectral response from 900 to 1700 nm. Each pixel has a size of 20 x 20 μm. The detector is Peltier-cooled to -80°C to minimize thermally generated noise and, thus improve signal-to-noise ratio. The NIRvana 640 camera has a 16-bit digitization, with low readout noise. Other components of the instrument are a SWIR wide-angle primary lens (~120° field of view), an f/1.0 SWIR-optimized telecentric optical system, a thermally stabilized 8-position filter wheel with four narrow-band (three-cavity) interference filters, brushless DC servo motor drive for the filter wheel, filter wheel control unit, power supply and, instrument control and data acquisition computer. As the center wavelengths of interference filters are temperature dependent, the filter wheel's closed-loop temperature controller heats the filter wheel cavity to +23±0.1°C. Along with four narrow-band interference filters, a blocking dark filter is used to collect data for subtraction of dark-current noise from nighttime measurements. The blocking dark filter's emissivity is matched to that of the narrow-band interference filters thus ensuring the dark-current noise collected with it is representative of the dark-current noise present in nighttime measurements. The primary wide-angle lens, telecentric optics and re-imaging lens were custom designed and produced for measurements in the SWIR. The re-imaging optics consist of a doublet field lens and a combination of a second doublet and a f/1 compound lens in front of the sensor. The optical design of the OH imager inscribes the 120° field of view image circle within the short (vertical) dimension of the 640 x 512 pixel sensor, resulting in an image circle with a 512 pixel diameter. The image is therefore cropped to 512 x 512 pixels for image processing purposes. The OH imager was designed and built by Keo Scientific Ltd (http://keoscientific.com). The main instrument characteristics are summarized in Table 1.

Nightglow observations are restricted to nighttime and twilight (solar elevation angle less than -6.5°) for optimum signal quality. At the arctic location of Kiruna, we are thus able to conduct nightglow measurements from September until middle of April. The exposure times for the OH, background and dark filters are set to 30 s, whereas the exposure time for the $O_2$ IR A-band filter is set to 40 s, yielding a complete filter cycle in 2.75 min. This yields the Nyquist frequency of the

instrument equal to 0.019 s$^{-1}$ (5.5 min). To avoid aliasing with the Brunt-Väisälä frequency in the mesopause which is varied around 0.021 s$^{-1}$ (5 min), the instrument will be used to resolve the high frequency range in the gravity wave spectrum, corresponding to observed wave periods of about 10 minutes and more. Figure 1 shows the OH imager installed at the Knutstorp observatory (67.86°N, 20.42°E), 2 km north of IRF, on 29 November 2022.

## 3 Temperature estimation and its uncertainty, error analysis, calibration technique

### 3.1 Temperature estimation

The temperature of the air in the mesopause region is determined using the brightness ratio of the two vibrational-rotational lines P$_1$(2) and P$_1$(4) in the OH (3-1) band. This method is based on the assumption that the rotational level populations are in local thermodynamic equilibrium (LTE). Although the OH vibrational states are excited due to non-thermal processes, the rotational level population is still in LTE due to the long radiative lifetimes of vibrational states (e.g., Savigny, 2017). This method has been proved to be valid and has been applied for several decades in temperature measurements of the mesopause region (Sivjee and Hamwey, 1987; Makhlouf et al., 1995; Espy and Stegman, 2002; Pertsev and Perminov, 2008; Suzuki et al., 2008; Offermann et al., 2010; Grygalashvyly et al., 2014; Pautet et al., 2014; Kalicinsky et al., 2016; Perminov et al., 2018).

For the OH rotational temperature estimation, we use Eq. (4) from Pautet et al. (2014). At the same time, we have corrected this equation for newly estimated Einstein A-coefficients for the OH (3-1) band at the P$_1$(2) and P$_1$(4) lines as calculated by Brooke et al. (2016). New A-coefficients for the P$_1$(2) and P$_1$(4) lines are 9.895802 and 13.15222 s$^{-1}$, respectively. After the correction for the A-coefficients, the following equation for the OH rotational temperature estimation $T_r$ is obtained:

$$T_r = \frac{259.58}{\ln(2.658 \cdot R)} \tag{1}$$

where $R$ is the brightness ratio B(P$_1$(2))/B(P$_1$(4)) inferred from measurements of the P$_1$(2) and P$_1$(4) lines in the absolute units (Rayleigh). Equation (1) differs from Eq. (4) by Pautet et al. (2014) in terms of the multiplier at the $R$ parameter: 2.658 versus 2.644 used in Eq. (4) by Pautet et al. (2014). This results in 0.7-1.2 K temperature difference between temperature estimations in the present paper and Pautet et al. (2014) for a typical range of winter temperatures (180-240 K) in the mesopause region.

### 3.2 Absolute calibration

The purpose of the absolute calibration of the OH imager is to establish a relationship between the arbitrary, instrument-dependent raw pixel values (counts) and the column emission rate in units of the Rayleigh (R). The imager absolute calibration is a multi-stage process that was performed by Keo Scientific Ltd. First, filter bandpass curves for the three OH(3-1) filters (atmospheric background, P$_1$(2) and P$_1$(4) lines) were measured by using a tunable SWIR laser. Second, on-axis broadband sensitivity was determined by imaging the output of a calibrated (National Institute of Standards and Technology, NIST-traceable) integrating sphere through the imager, including the filters, and onto the InGaAs photodiode array sensor. Additionally, an on-axis 'at-wavelength' sensitivity was measured for each filter using a collimated laser beam calibrated with a (NIST-traceable) optical power meter through the

imager and onto the InGaAs photodiode array sensor. This resulted in the final absolute calibration coefficients, valid on-axis. Finally, a flat-field correction procedure was performed to compensate for off-axis vignetting, resulting in uniformity coefficients, one for each OH filter, allowing one to apply the absolute calibration coefficients to any pixel in the image. The flat-field correction consists of three steps: (1) the cylindrically symmetric component of the lenses is characterized first. This is done on an optical bench (180° rotary table) and an integrating sphere, where the sphere is illuminated with a tunable laser at the emission wavelengths of each of the two OH channels; (2) the imager is inserted into the integrating sphere and an additional diffuser is placed over the fisheye lens. Images are taken with the filters in place and with the uncoated glass blank in place. This is to find the attenuation of the filters relative to the uncoated glass blanks; (3) Zernike polynomials are then used for the fitting to produce the smooth flat fields. A calibration certificate with supporting files was issued by Keo Scientific.

### 3.3 Geometrical calibration

The geometrical calibration of the OH imager is a necessary procedure to measure absolute horizontal coordinates of each pixel of the sensor at the instrument location. The geometrical calibration was performed by analyzing images of a clear night sky with reference stars by using a wide-band filter (1000-1600 nm), also present in the filter wheel. The PPM Star Catalogue (Positions and Proper Motions Star Catalogue) was used to identify positions of the reference stars. The camera optical model was chosen to be described by the 3rd order polynomial $P$:

$$P=a_1 \cdot x^3 + a_2 \cdot x^2 y + a_3 \cdot y^2 x + a_4 \cdot y^3 + a_5 \cdot x^2 + a_6 \cdot xy + a_7 \cdot y^2 + a_8 \cdot x + a_9 \cdot y + a_{10} \qquad (2)$$

where ten $a_i$ parameters are free coefficients, $x$ and $y$ are horizontal coordinates of the reference stars on the analyzed image. By comparing theoretical horizontal coordinates of reference stars (more than 50 reference stars have been identified in reference images) with their measured coordinates, ten free coefficients of the 3rd order polynomial were calculated in the least-squared sense. These ten coefficients describe all possible optical distortions in the whole camera optical path. This procedure finally resulted in calculating absolute horizontal coordinates (elevation and azimuth angles) of each pixel followed by a georeference procedure to project each pixel onto the Earth's surface. For this, the mean altitudes of the OH and $O_2$ layers were chosen as 87 km and 94 km, respectively. The spatial horizontal resolution in the center of the image is about 0.3 km, reaching ~1.2 km at the edges of the image.

### 3.4 Error analysis

In order to estimate the total error of temperature estimation, we need to rewrite the basic Eq. (1). The $R$ brightness ratio in Eq. (1) depends on four variables represented by their raw measurement values in counts: raw intensity of the $P_1(2)$ line $I_{P12}$, raw intensity of the $P_1(4)$ line $I_{P14}$, atmospheric background raw intensity $I_{bg}$ and instrumental dark-current noise $n_{dc}$. The raw intensities $I_{P12}$ and $I_{P14}$ mean that no corrections for the atmospheric background and noise have been made. Then Eq. (1) can be rewritten in terms of these four variables as follows:

$$T_r = \frac{259.58}{\ln\left(2.658 \cdot \dfrac{k_1 \cdot k_4 \cdot (k_2 \cdot I_{P12} - k_3 \cdot I_{bg} + 0.22 \cdot n_{dc})}{k_5 \cdot k_7 \cdot (k_6 \cdot I_{P14} - k_3 \cdot I_{bg} + 0.60 \cdot n_{dc})}\right)} \qquad (3)$$

where $k_1$, $k_2$, $k_3$, $k_4$, $k_5$, $k_6$, $k_7$ are the coefficients determined in a laboratory during the absolute calibration procedure. Equation 3 comes from Eq. 1 in the following way. The $R$ brightness ratio in Eq.1 is in the absolute units (Rayleigh). The instrument registers emission intensities (P$_1$(2), P$_1$(4) and atmospheric background) in relative digital units (counts). In order to relate relative to absolute units, the absolute calibration is performed. The main part of this procedure is to determine filter absolute sensitivities which are different for each filter. These are the coefficients $k_2$, $k_3$ and $k_6$ in Eq.3. In addition, the dark noise is subtracted both from the P$_1$(2) and P$_1$(4) lines as well as from the atmospheric background line which, in turn, is finally subtracted from the P$_1$(2) and P$_1$(4) lines. Since the coefficients $k_2$, $k_3$ and $k_6$ are different this procedure results in different constants (0.22 and 0.60) for the subtracted dark noise in the numerator and denominator in Eq.3. These constants have positive signs since the dark noise is subtracted from the P$_1$(2) and P$_1$(4) lines as well as from the atmospheric background line having different coefficients. The coefficients $k_4$ and $k_7$ describe the flat field correction factors being different for each OH (3-1) emission line. Finally, the coefficients $k_1$ and $k_5$ convert photometric units to the Rayleigh, which includes several multiplies and the geometric etendue.

In order to estimate these four raw measurement values ($I_{P12}$, $I_{P14}$, $I_{bg}$, $n_{dc}$) and their errors, we follow the method proposed by Pautet et al. (2014). Measurements should be performed during a time when atmospheric and geomagnetic perturbations are at a minimum (i.e., small variations in raw OH emission lines and in the atmospheric background, no contamination by tropospheric clouds, no auroral emissions). A period of one hour on 6 April was chosen to evaluate the four raw measurement variables and their errors. Figure 2 illustrates the raw intensities for the P$_1$(2), P$_1$(4) lines as well as for the raw atmospheric background (BG) measured at the zenith over a single-pixel area (0.003 x 0.003°). To reduce measurement perturbations as much as possible, the raw data were smoothed by a 3-point moving average. Then the residuals were determined by subtracting the smoothed curves from the raw intensity data and their standard deviations were estimated. There have been obtained the following smoothed average values and their standard deviations: 57,711±290 counts for the P$_1$(2) line, 48,552±110 counts for the P$_1$(4) line, and 31,301±126 counts for the BG. The measured random temperature error ($\delta T_m$) is calculated by using a general equation of error propagation in the case of a function having several variables (Taylor, 1997):

$$\delta T_m = \sqrt{\left(\frac{\partial T}{\partial I_{P12}} \delta I_{P12}\right)^2 + \left(\frac{\partial T}{\partial I_{P14}} \delta I_{P14}\right)^2 + \left(\frac{\partial T}{\partial I_{bg}} \delta I_{bg}\right)^2} \qquad (4)$$

where $\partial T/\partial I_{P12}$, $\partial T/\partial I_{P14}$ and $\partial T/\partial I_{bg}$ are the partial derivatives of $T_r$ (see Eq. 3) with respect to raw intensity values P$_1$(2), P$_1$(4) and atmospheric background, respectively, and $\delta I_{P12}$, $\delta I_{P14}$ and $\delta I_{bg}$ are their standard deviations. When inserting the parameters that were estimated above into Eq.4, a temperature measurement random error of 1.80 K is derived.

The temperature instrumental error is mainly governed by the dark-current noise of the CCD sensor as well as by the coefficients $k_{1,2,3,4,5,6,7}$ in Eq. 3. The dark-current noise was continuously monitored by using the dark blocking filter as part of the measurement cycle (every 2.75 min), and was subsequently subtracted from the recorded raw intensity values of the OH and $O_2$ emission lines. Example of the dark-current noise is shown in Fig. 3, demonstrating a linear stability of the dark-current noise during the night of 6-7 April 2023, with the small statistically insignificant regression coefficient equal to 186±355 counts/day. The mean value of the dark-current noise $n_{dc}$ and its standard deviation $\delta n_{dc}$ is 30,900±183 counts for this night. The relative errors of the coefficients $k_{1,2,3,4,5,6,7}$ were estimated to vary from 0.8% to 4.2% in the course of the absolute calibration that results in the temperature errors ranging from 0.1 K to 2.1 K.

The total (instrumental + measurement) error $\delta T_{tot}$ is calculated by adding the terms for the instrumental error to Eq.4:

$$\delta T_{tot} = \sqrt{\left(\frac{\partial T}{\partial n_{dc}}\delta n_{dc}\right)^2 + \left(\frac{\partial T}{\partial k_i}\delta k_i\right)^2 + \left(\frac{\partial T}{\partial I_{P12}}\delta I_{P12}\right)^2 + \left(\frac{\partial T}{\partial I_{P14}}\delta I_{P14}\right)^2 + \left(\frac{\partial T}{\partial I_{bg}}\delta I_{bg}\right)^2} \quad (5)$$

where $\partial T/\partial n_{dc}$ and $\partial T/\partial k_{i=1,2,3,4,5,6,7}$ are partial derivatives of the temperature with respect to dark-current noise $n_{dc}$ and the coefficients $k_{i=1,2,3,4,5,6,7}$, $\delta n_{dc}$ and $\delta k_{i=1,2,3,4,5,6,7}$ are their standard deviations. Based on the above estimated mean dark-current noise and its standard deviation, the mean temperature instrumental error due to the dark-current noise is equal to 0.15 K on the night 6-7 April 2023. The instrumental temperature error due to the coefficients $k_{1,2,3,4,5,6,7}$ is equal to 3.37 K, and the total instrumental temperature error ($\sqrt{(0.15^2 + 3.37^2)}$) is 3.38 K. Finally, the resulting rotational temperature and its total error ($\sqrt{(1.80^2 + 3.38^2)}$) is equal to 193.9±3.8 K for this particular case shown in Fig. 2.

At the same time, we should note that actual variability of atmospheric conditions due to complex wave dynamics (turbulence, gravity and planetary waves, solar and lunar tides, seasonal variations) is greater resulting in an average standard deviation of the mesopause temperature above Kiruna of ~10 K for the 2023 (see section 5). The most important is that the OH imager enables resolving small temperature variations of 4-6 K due to small-scale gravity waves as will be demonstrated in section 5.

## 4 Temperature validation and comparison

### 4.1 OH imager and lidar measurements

A Rayleigh/Mie/Raman backscatter lidar located at Esrange (~40 km eastward of Kiruna) is used to validate the OH (3-1) temperatures as measured by the OH imager. The Esrange lidar was developed by the Bonn University to monitor aerosols in the troposphere, stratosphere and mesosphere (Blum and Fricke, 2005). The signal from the aerosol-free part of the atmosphere can be used to determine the temperature assuming atmospheric hydrostatic equilibrium. The vertical and time resolutions of the obtained measurements are 150 m and 4.2 min, respectively. We used lidar backscattered signal from the 532-nm wavelength channel to calculate an atmospheric temperature profile above Esrange. Note that a temperature profile measured by the Esrange lidar is located in the field of view of the OH

imager, i.e., common volume simultaneous temperature measurements have been conducted to validate the OH (3-1) temperatures.

The Esrange lidar was operated when weather conditions were favorable during 7 nights in the period from January to March 2023. An example of a temperature comparison, made on the night 2/3
February 2023, between the OH imager and lidar is shown in Fig. 4. The OH temperature measurement has been selected from the temperature map having the closest position to Esrange with 0.3 km uncertainty. At the upper end of the profile (at around 90 km) a seed temperature equal to 188 K was taken from Aura/MLS satellite temperature measurements. Mean values of the OH (3-1) and lidar temperatures averaged for the time interval between 18:43 and 05:58 UT on this night are presented. In
addition, the lidar temperature profile has been smoothed by using a 1.5 km running average in height in order to reduce large fluctuations in the upper part of the profile. The Aura/MLS temperature profile, having the closest position to Kiruna (about 420 km away) for this case, is shown in Fig. 4 by the red line as well. Since the OH (3-1) temperature is measured across the whole OH layer between about 82 and 92 km, with the mean height of ~87 km, for validation purposes the lidar temperature profile has
been processed using a Gaussian function with the maximum at 87 km and having the Full Width at Half Maximum (FWHM) of 9.3 km height-weighted between 82 and 90 km; the same technique as was used by Pautet et al. (2014). The average height-weighted lidar temperature is equal to 191.8±12.3 K and the average OH (3-1) temperature is 193.6±8.3 K. The temperature difference between these estimations is 1.8 K for this particular night, i.e., within the error margin. Following the same
technique, we have made comparisons between the OH (3-1) and lidar temperatures for all available 7 nights of simultaneous common-volume measurements, resulting in the following statistics: average temperature difference, including sign, between the OH imager and Esrange lidar is -0.2±1.6 K. It means that, from statistical point of view, the difference between the OH (3-1) and Esrange lidar temperatures is about zero. The maximum temperature difference between these instruments was -2.9
K on the night 5-6 March 2023. Note that we estimate the Gaussian height-weighted lidar temperature between 82 and 90 km with the mean height of 87 km. However, the actual height distribution of the OH layer above Esrange is unknown since it varies, in general, in time and space (i.e., Pautet et al., 2014). Thus, one can conclude that there is a good agreement between the OH (3-1) and Esrange lidar temperature measurements on individual nights (within 3 K) as well as on an average basis (close to
zero), after taking into account dynamical processes in the mesopause region and instrument errors.

### 4.2 OH imager and Aura/MLS measurements

The Microwave Limb Sounder (MLS) radiometer onboard the Aura satellite provides temperature profiles of the middle atmosphere with near-global coverage. Aura/MLS temperature data (ver.5.0 and
level 2 data quality) can be obtained from the NASA public web-site: https://acdisc.gesdisc.eosdis.nasa.gov/data/Aura_MLS_Level2/. The description of the MLS temperature retrieval and its validation can be found in Froidevaux et al. (2006) and Schwartz et al. (2008). We used Aura/MLS temperature profiles as an additional way to compare temperatures that were retrieved with the OH imager.

Vertical resolution of Aura/MLS temperature measurements is about 11-12 km in the mesopause

region and temperature precision of individual profiles is about 3.5 K at mesopause heights (see the

Aura/MLS data quality and description document available at the NASA web-site). Aura/MLS

temperatures at the pressure level of $4.6 \cdot 10^{-3}$ hPa (about 86 km altitude) have been chosen to be closest

to the OH imager location (less than 300 km away from Kiruna) and measured simultaneously with OH

(3-1) temperatures within the same hour. The comparison between OH imager and Aura/MLS

temperature measurements for the period January-April 2023 is illustrated in Fig. 5. In total, 22

individual temperature measurements have been collected. One can see that there are several cases in

which the temperatures agree well within their uncertainties (days of year 14, 58, 61, 65, 83, 85, 86, 92,

93, 99, 101). For other days the agreement is worse. Note that even if the Aura/MLS instrument

measures temperature in the closest proximity to the OH imager, the Aura/MLS scans horizontally a

rather big volume of the mesopause (about 250-280 km) along its trajectory which is not exactly the

same mesopause volume as the OH imager measures; besides, the vertical resolution of Aura/MLS

temperature data in the mesopause is not equal to the width of the OH layer. All these factors

complicate a precise comparison. Natural variabilities of the atmosphere due to wave dynamics and

chemical processes result in different temperatures in different atmospheric volumes. At the same time,

two important conclusions can be drawn from Fig. 5: a) there is no systematic bias (neither negative

nor positive) of the OH (3-1) temperature relative to Aura/MLS temperature since there are OH (3-1)

temperatures higher and lower Aura/MLS temperatures; b) the average temperature difference,

including sign, between the OH imager and Aura/MLS is 2.8±7.8 K. This is a small difference taking

into account the above-mentioned limitations.

## 5 Results and discussion

    Maps of OH (3-1) intensities in the Rayleigh and $O_2$ IR A-band intensity in relative units as well as

the OH (3-1) temperature map on 16 February 2023 are illustrated in Fig. 6. One can see there is a fine

modulation of the temperature (Fig. 6d) due to small-scale gravity waves with wavelengths of 4-8 km,

overlying larger gravity wave crests; the temperature amplitudes due to these small-scale waves are in

range of 4-6 K. It means that this case clearly demonstrates the ability of the OH imager to resolve

small-amplitude small-scale dynamical temperature variations that is of importance for studying a high

frequency range of the gravity wave spectrum. Figure 7 illustrates a zoom of the images shown by the

black squares in Fig. 6, which focuses on the small-scale wave structure of about 4-6 km wavelength in

the $P_1(2)$, $P_1(4)$ and temperature maps. Residuals of the $P_1(2)$ and $P_1(4)$ intensities are shown after

subtraction of the mean intensity values from these zoom images. Note that the temperature estimation

(see Eq. 1) is not a linear function of the $P_1(2)$ and $P_1(4)$ line intensities, meaning that the linear scale of

the $P_1(2)$ and $P_1(4)$ changes, shown in Fig. 7a,b, cannot directly represent the temperature changes in

the observed small-scale wave structures shown in Fig. 7d. Another important feature of this case there

is a medium-scale modulation, oriented in the direction from north-west to south-east, having

horizontal wavelengths of 20-40 km and temperature amplitudes of 7-10 K. These medium-scale

gravity wave crests are clearly seen on all the four maps, meaning that the same gravity wave package

was propagating through both the OH layer (maximum is at ~87 km) and the $O_2$ layer (maximum is at

~94 km). An analysis of a sequence of temperature maps have demonstrated that these medium-scale gravity waves were moving from south-west to north-east, with the observed horizontal phase speed of about 30-35 m/s. This type of medium-scale gravity waves is common in the mesopause region (e.g., Pautet et al., 2011; Demissie et al., 2014). Thus, the OH imager allows retrieving information on gravity waves propagating both in horizontal and vertical domains in the mesopause region. A detailed

analysis of gravity wave characteristics and of the gravity wave spectrum is beyond the scope of this paper and will be addressed in future studies. A file named "OH_imager_video_160223.avi" contains a video sequence of the intensities and temperature maps shown in Fig. 6, which can be found in the supplementary material to the paper (see Data Availability**)**. The video demonstrates a motion of atmospheric gravity waves of various scales, preferentially moving from the south-west to the north-

east.

    The OH (3-1) daily mean rotational temperatures measured above Kiruna for the 2023 winter season (13 January-16 April) are illustrated in Fig. 8 by the black circles. The OH (3-1) temperature values are within the range of 178-224 K. The average daily winter temperature for this particular period is equal to 203±10 K. For comparison, Aura/MLS temperatures, close to Kiruna location (less

than 675 km away), are indicated by the red circles. There is a good agreement between these time series, with a small average temperature difference of 1.9±6.9 K. The temperature behavior agrees well with a general seasonal temperature behavior in the winter mesopause having a maximum in the period of December-February followed by a rapid temperature decrease in March-April (e.g., Perminov et al., 2018). Kim et al. (2017) have analyzed mesopause temperatures from OH airglow measurements with

a Fourier Transform Spectrometer (FTS) for winter periods of 2003-2014 at Kiruna. The authors have found variations in the winter mesopause temperature above Kiruna being in the range of 180-230 K. This temperature range agrees well with the OH (3-1) temperature values presented in Fig. 8. Cho et al. (2011) have studied OH and $O_2$ airglow temperatures for winter periods of 2001-2007 measured with a Spectral Airglow Temperature Imager (SATI) instrument and a Michelson interferometer at high

northern latitudes at Resolute Bay and Esrange, respectively. Cho et al. (2011) have demonstrated winter mesopause temperatures to vary between 180 and 245 K that again agrees well with the temperature range obtained by the OH imager. Sigernes et al. (2003) have studied daily mesospheric winter temperatures for more than 20 years of ground-based spectral measurements of the OH layer over Svalbard from 1980 to 2001. The authors have found the average daily winter temperature to be

equal to 208±15 K, with the maximum and minimum temperatures of 257 K and 168 K, respectively, emphasizing that the mesospheric temperature variations over Svalbard in winter are extremely high. The average daily winter OH (3-1) temperature that was estimated in the present study is close to the value found by Sigernes et al. (2003), taking into account large temperature deviations of the average daily values found in both papers.

It is interesting to discuss the following feature in the OH (3-1) temperature behavior. There have been observed two rather strong temperature enhancements on day 52 (21 February 2023) and on day 66 (7 March 2023). The 2023 Sudden Stratospheric Warming (SSW) event started around 1st February 2023 (https://www.climate.gov/news-features/event-tracker/disrupted-polar-vortex-brings-sudden-stratospheric-warming-february). It is known that a SSW is accompanied by both cooling and warming

of the mesopause region: cooling is observed at the time of a SSW onset or a few days later while a warming is observed with a delay of about 20-25 days (Cho et al., 2004; Shepherd et al., 2014). Sometimes several mesopause warmings (wavy oscillations) can be generated after a SSW onset (see Fig. 1b,f in Shepherd et al., 2014). This is exactly what was observed in the OH (3-1) temperature behavior: first, there was the OH (3-1) temperature decrease between days 32-36 (close to the 2023

SSW onset), then the two strong temperature maxima occurred on days 52 and 66, i.e., about 20 and 34 days after the 2023 SSW onset. A detailed link of the OH (3-1) temperature above Kiruna to SSW events will be investigated in future papers.

The two emission layers (OH and $O_2$) are varied in space and time, making different height distances between these layers. This complicates an analysis dealing with a vertical propagation of

gravity waves. At the same time, if the same wave package, having the same horizontal wavelength and observed phase velocity as well as propagation direction, is observed both in the OH and $O_2$ layers, one can assume that the same gravity wave was propagating both in horizontal and vertical domains. According to the general theory of gravity waves (e.g., Gossard and Hook, 1975) a gravity wave propagates at some angle to the vertical, with tilted phase lines. This should result in an observed phase

shift of the same gravity wave between the OH and $O_2$ layers. Once a phase shift and horizontal wavelength are estimated from the OH and $O_2$ maps, one can calculate the vertical wavelength by using the following relation:

$$\lambda_z = \lambda_x / \tan(\alpha) \qquad (6)$$

where $\lambda_z$ and $\lambda_x$ are the vertical and horizontal wavelengths of a gravity wave, $\alpha$ is the angle between

wave phase lines and the vertical. Furthermore, if the buoyancy frequency $N$ is a known quantity or is estimated by using lidar or satellite temperature profiles, one can deduce the intrinsic frequency $\omega$ of a gravity wave from the following relation:

$$\omega = \pm N \cdot \cos(\alpha) \qquad (7)$$

Substituting known values of $\omega$, $N$ and $\lambda_x$ into the dispersion relation for gravity waves one can

estimate a vertical wavelength again, thus verifying the first estimation of a vertical wavelength. Note that this method is valid for a limited number of gravity waves having vertical wavelengths less that the height distance between the two layers (about 7 km).

Another simple method of the estimation of a vertical wavelength of a gravity wave is based on the assumption that the height difference $D$ between the two layers is a known quantity (Fagundes et al.,

1995; Schmidt et al., 2018). If a horizontal phase shift $\Delta\varphi$ of a considered wave package between the both layers is estimated then one can calculate its vertical wavelength $\lambda_z$ using the following relation:

$$\lambda_z = D \cdot 2\pi / \Delta\varphi \qquad (8)$$

We will use the both approaches to estimate vertical wavelengths of gravity waves propagating through the OH and $O_2$ layers, which will be supported by model studies of gravity waves propagating through

the whole atmosphere up to the mesopause level (Dalin et al., 2015; Dalin et al., 2016).

**6 Conclusions**

A novel infrared imager (OH imager) began operating in November 2022 in Kiruna (northern Sweden), aiming at studies of nightglow emissions coming from hydroxyl and molecular oxygen layers

in the mesopause region. The OH imager registers selected emissions lines (1523 and 1542 nm) in the OH (3-1) band to obtain emission intensity and temperature maps at around 87 km altitude. Also, the imager records infrared emissions coming from the $O_2$ IR A-band airglow at 1269 nm in order to obtain $O_2$ emission intensity maps at a slightly higher altitude at ~94 km. This technique allows tracing gravity wave disturbances both in horizontal and vertical domains in the mesopause region. The lee location of

the OH imager favors studies of mountain waves, their probable influence on the mesopause temperature and dynamics.

  Temperature maps demonstrate the ability of the OH imager to resolve small-scale (4-8 km) small-amplitude (4-6 K) gravity waves as well as medium-scale gravity waves having horizontal wavelengths of 20-40 km and temperature amplitudes of 7-10 K. Such medium-scale waves have propagated

through both the OH and $O_2$ layers.

  The comparison between the OH imager and collocated Esrange lidar shows a good agreement between OH (3-1) and Esrange lidar temperature measurements on individual nights (within 3 K) as well as on an average basis (-0.2±1.6 K). The comparison between the OH imager and Aura/MLS radiometer demonstrates no systematic temperature bias between these instruments as well as small

average temperature difference of 2.8±7.8 K, which is likely to be caused by the variable state of the mesopause temperature over spatial scales of their different observational volumes.

  OH (3-1) daily mean rotational temperatures measured above Kiruna for the 2023 winter season (January-April) demonstrate typical seasonal variations with a maximum in February and a rapid temperature decrease in April in the winter polar mesopause. The OH (3-1) temperature values are

within the range of 178-224 K that agrees well with other high latitude temperature measurements in the winter mesopause. The average daily winter temperature for the considered period is 203±10 K.

**Code availability**

Matlab codes used in this study for the OH image processing are available upon request from the

corresponding author (pdalin@irf.se).

**Data availability**

Processed OH image data published in the paper's figures are available via the Swedish National Data Center (https://snd.gu.se/en/catalogue/dataset/preview/d5776ec9-d346-4e6a-a8c3-1445e75201b6/1).

Underlying research data are available upon request to Peter Dalin (pdalin@irf.se). Once the commissioning period of the OH imager instrument is completed, registrations will be made freely available for non-commercial scientific usage in accordance with other data obtained from the Kiruna Atmospheric and Geophysical Observatory (KAGO) at the Swedish Institute of Space Physics (IRF). Aura/MLS temperature data (ver.5.0 and level 2 data quality) can be obtained from the NASA public

web-site: https://acdisc.gesdisc.eosdis.nasa.gov/data/Aura_MLS_Level2/. Esrange lidar data can be made available upon request to Peter Dalin (pdalin@irf.se) or Jonas Hedin (jhedi@misu.su.se).

**Author contributions**

Author contributions. PD: conceptualization, methodology, installation of the OH imager, data processing and validation, matlab and python codes, statistical analysis, data visualization and archiving, optical calibration, writing the original manuscript. UB: conceptualization, writing a proposal to the Swedish Research Council (Vetenskapsrådet), installation of the OH imager, providing data storage. JK: conceptualization, writing a proposal to the Swedish Research Council (Vetenskapsrådet). PV: installation of the OH imager, software development, lidar operation. TN: conceptualization, installation of the OH imager, writing python codes. TT, DW and CU: methodology, design, manufacturing and absolute calibration of the OH imager. VP and NP: conceptualization, methodology, assisted with the interpretation of scientific results. JH: Esrange lidar preparation and operation. All co-authors contributed to writing and reviewing the paper.

## Competing interests

The authors declare that they have no conflict of interest.

## Acknowledgements

We are thankful to Daria Mikhaylova and Martin Rönnfalk (IRF-Kiruna) for their help with the installation of the OH imager.

## Financial support

The OH imager installation was financed and performed as part of the research infrastructure grant 2021-00360 from the Swedish Research Council (Vetenskapsrådet) to the Kiruna Atmospheric and Geophysical Observatory (KAGO) at the Swedish Institute of Space Physics (IRF).

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

**Table 1**. The main characteristics of the OH imager in Kiruna.

| Parameter | Value |
|---|---|
| CCD image sensor | InGaAs focal plane array |
| CCD format/ used for image processing | 640 x 512 / 512 x 512 pixels |
| CCD image area | 12.80 x 10.24 mm |
| Pixel size | 20 x 20 μm |
| Pixel full well high capacity at 2 MHz | 677,800 e- |
| Quantum efficiency | ~85% |
| Dark current at 2 MHz at -80°C | 21.1 e-/pix/sec |
| Temperature of detector using a Peltier cooler | −80°C at ambient temperature 21°C |
| Digitization | 16 bit |
| Field of view | 120° |
| Filters diameter | 76.2 mm |
| Narrow-band three-cavity filters centered at: | |
| OH(3-1) background ($\Delta\lambda$ FWHM) | 1521.0 nm ($\Delta\lambda$=2.2 nm) |
| OH(3-1) P1(2) line ($\Delta\lambda$ FWHM) | 1523.7 nm ($\Delta\lambda$=2.2 nm) |
| OH(3-1) P1(4) line ($\Delta\lambda$ FWHM) | 1542.8 nm ($\Delta\lambda$=2.1 nm) |
| $O_2$ IR A-band line | 1268.7 nm ($\Delta\lambda$=2.9 nm) |
| Exposure times for OH and $O_2$ filters | 30 and 40 s |
| Time resolution of the whole cycle | 165 s |
| Spatial resolution of OH measurements at 87 km | 0.3 km in the image center |
| Horizontal coverage at 87 km | 300 x 300 km |

**Figures:**

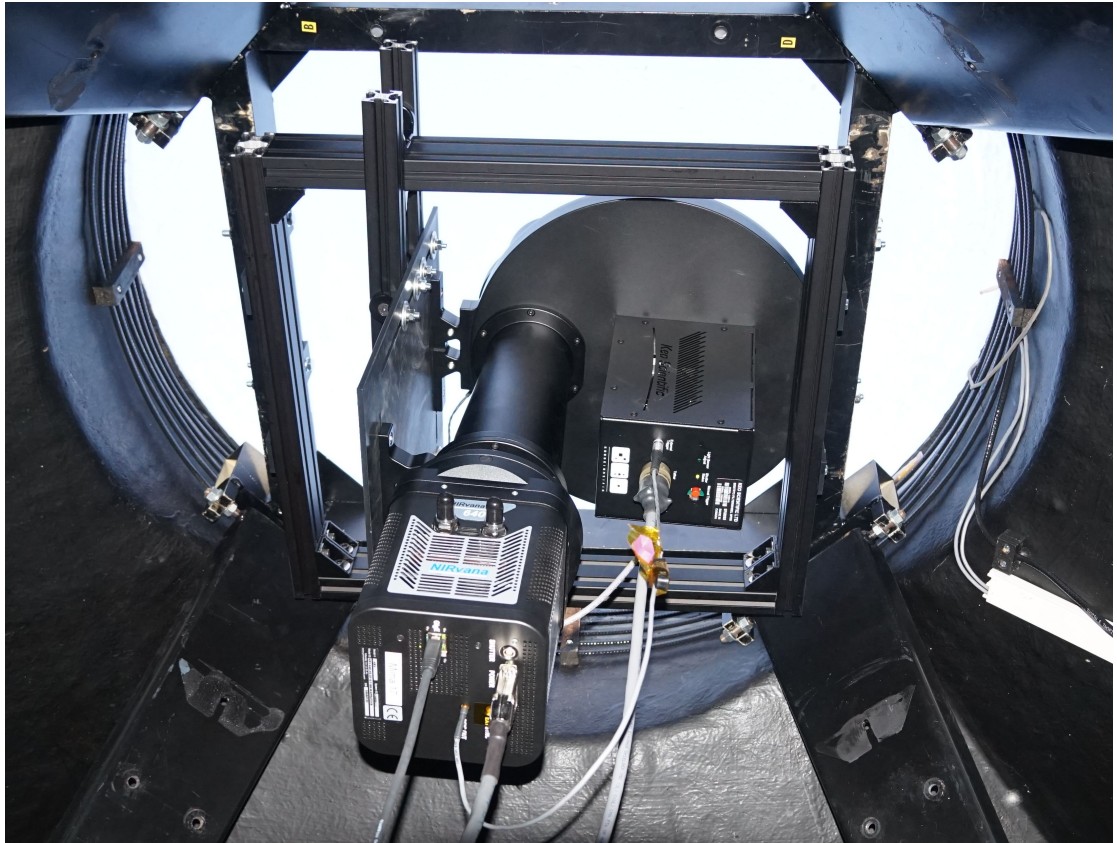

**Figure 1: The Keo Scientific OH imager installed in November 2022 at the Swedish Institute of Space Physics (IRF), located in Kiruna, Sweden (67.86°N, 20.42°E). The image was taken by Peter Dalin.**


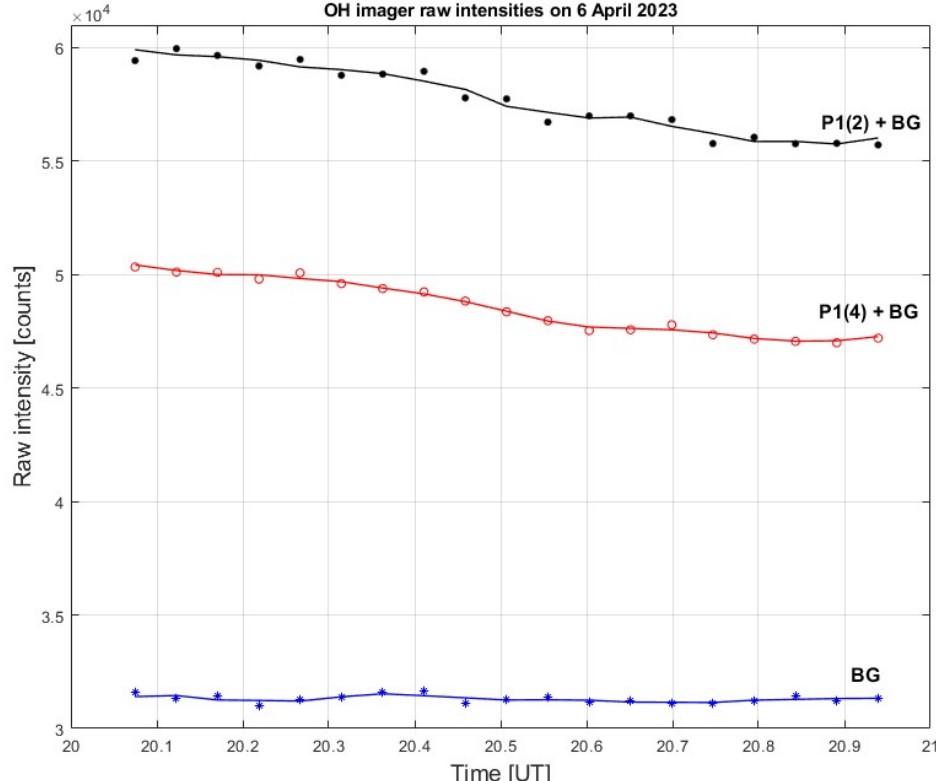

**Figure 2: Raw intensities of emission lines at the zenith as measured by the OH imager in Kiruna on 6 April 2023. The black dots, red circles and blue asterisks are raw intensities of the OH $P_1(2)$, $P_1(4)$ lines (including atmospheric background and noise level of the detector) and raw atmospheric background (BG), respectively. The black, red and blue lines are 3-point moving averages.**


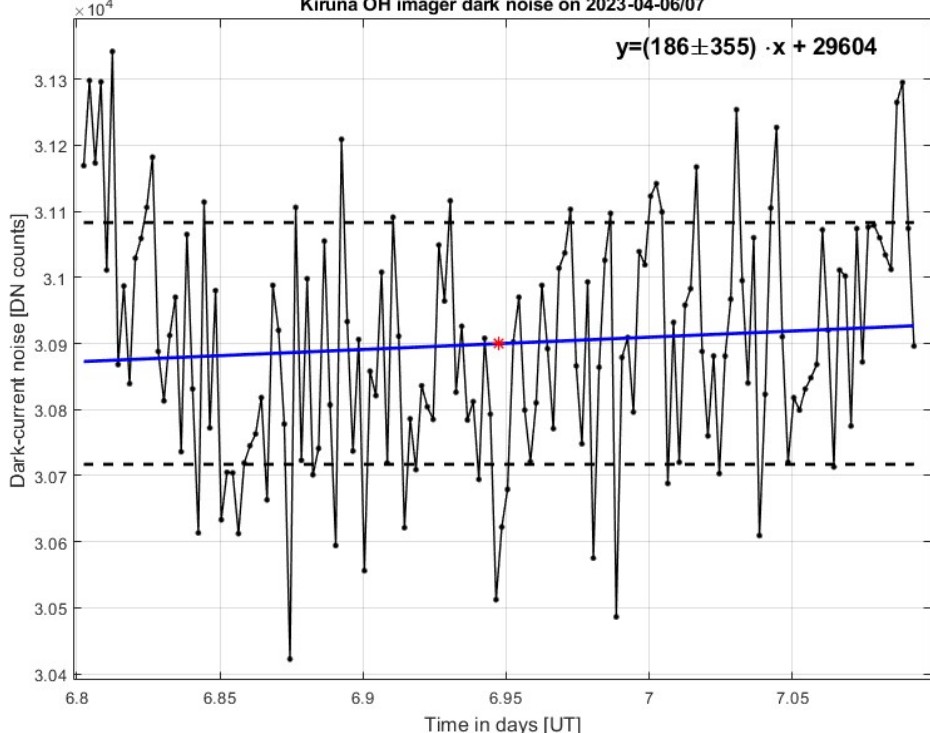

**Figure 3: Example of the dark-current noise of the OH imager in Kiruna on the night of 6-7 April 2023. The black dots are dark-current noise counts in the detector center, the blue line is the linear regression, the red asterisk is the mean dark-current noise and the dashed lines are one standard deviation around the mean noise.**


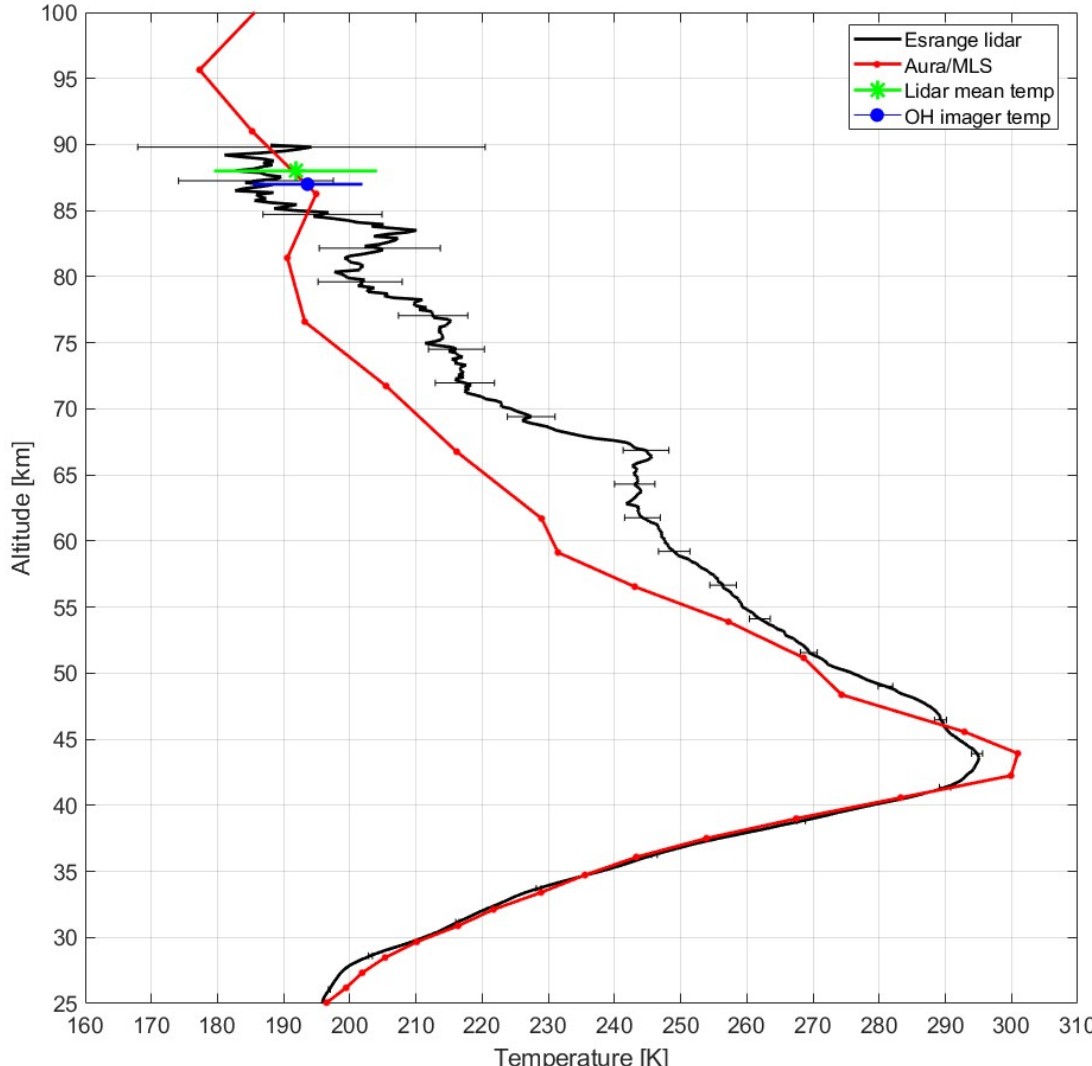

**Figure 4: Temperature measurements made between 18:43 and 05:58 UT on the night 2/3**
**February 2023. The blue dot is the mean OH (3-1) temperature at 87 km averaged over the same**
**time period. The black line is the Esrange lidar mean temperature profile. The green asterisk is**
**the average height-weighted lidar temperature as was calculated by vertical averaging across the**
**OH layer (see the text). Error bars for the OH imager and lidar measurements are one standard**
**deviation of the temperature variations for the night. The red line is the Aura/MLS temperature**
**profile which was chosen at 70.3°N 12.2°E, closest to the position of Kiruna (about 420 km away**
**from Kiruna), measured at 02:32 UT on 3 February 2023.**

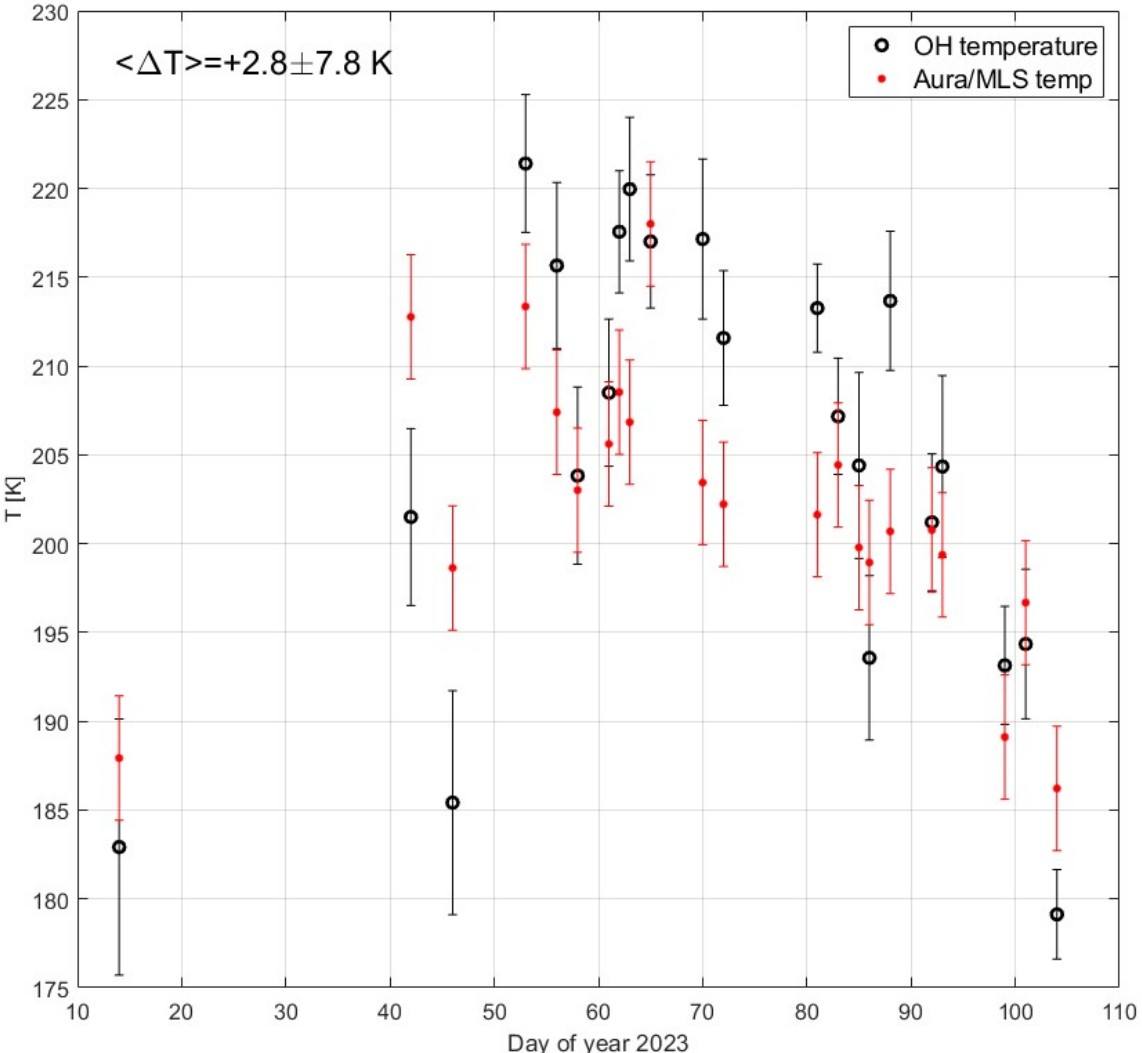

**Figure 5: A comparison between simultaneous measurements of the mesopause temperature performed by the OH imager (black circles) above Kiruna at ~87 km and Aura/MLS (red points) at 0.0046 hPa, ~86 km, close to the Kiruna location (distance difference is less than 300 km). Error bars of the OH (3-1) temperature are one standard deviations for a particular hour. Error bars of the Aura/MLS temperature are equal to 3.5 K (see the text).**



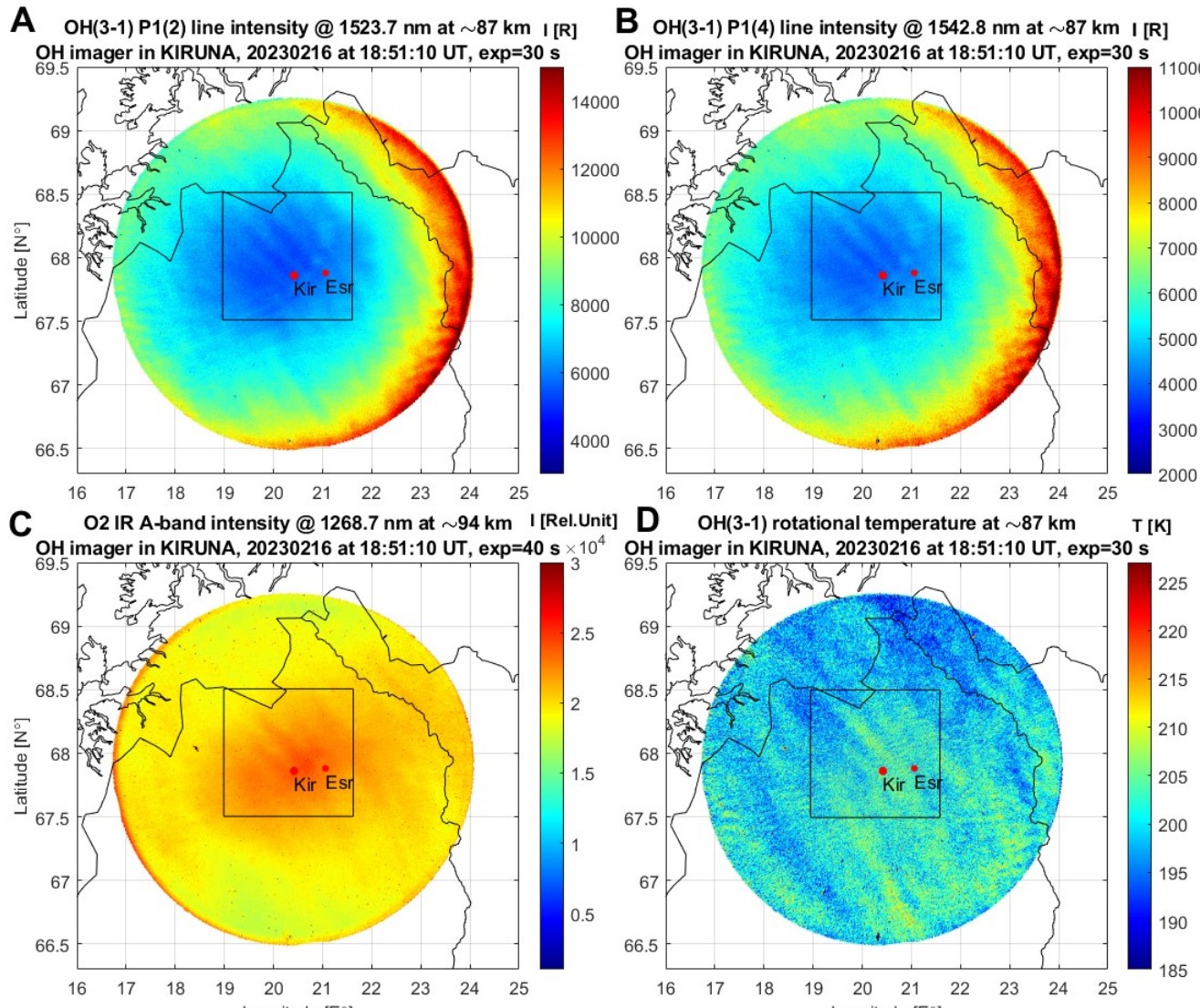

**Figure 6: Maps of airglow intensities and temperature obtained by the OH imager on 16 February 2023. (A) Intensity in the Rayleigh of the OH (3-1) P$_1$(2) line at 1523.7 nm. (B) Intensity in the Rayleigh of the OH (3-1) P$_1$(4) line at 1542.8 nm. (C) Intensity in relative units of the O$_2$ IR A-band at 1268.7 nm. (D) OH (3-1) rotational temperature estimated by the brightness ratio of the two OH (3-1) emission lines (see the text). The presented timestamp in all the maps corresponds to the time of the middle point of a current filter wheel cycle which takes 2.75 min. The black squares show a zoom of the images, which is shown in Fig.7.**

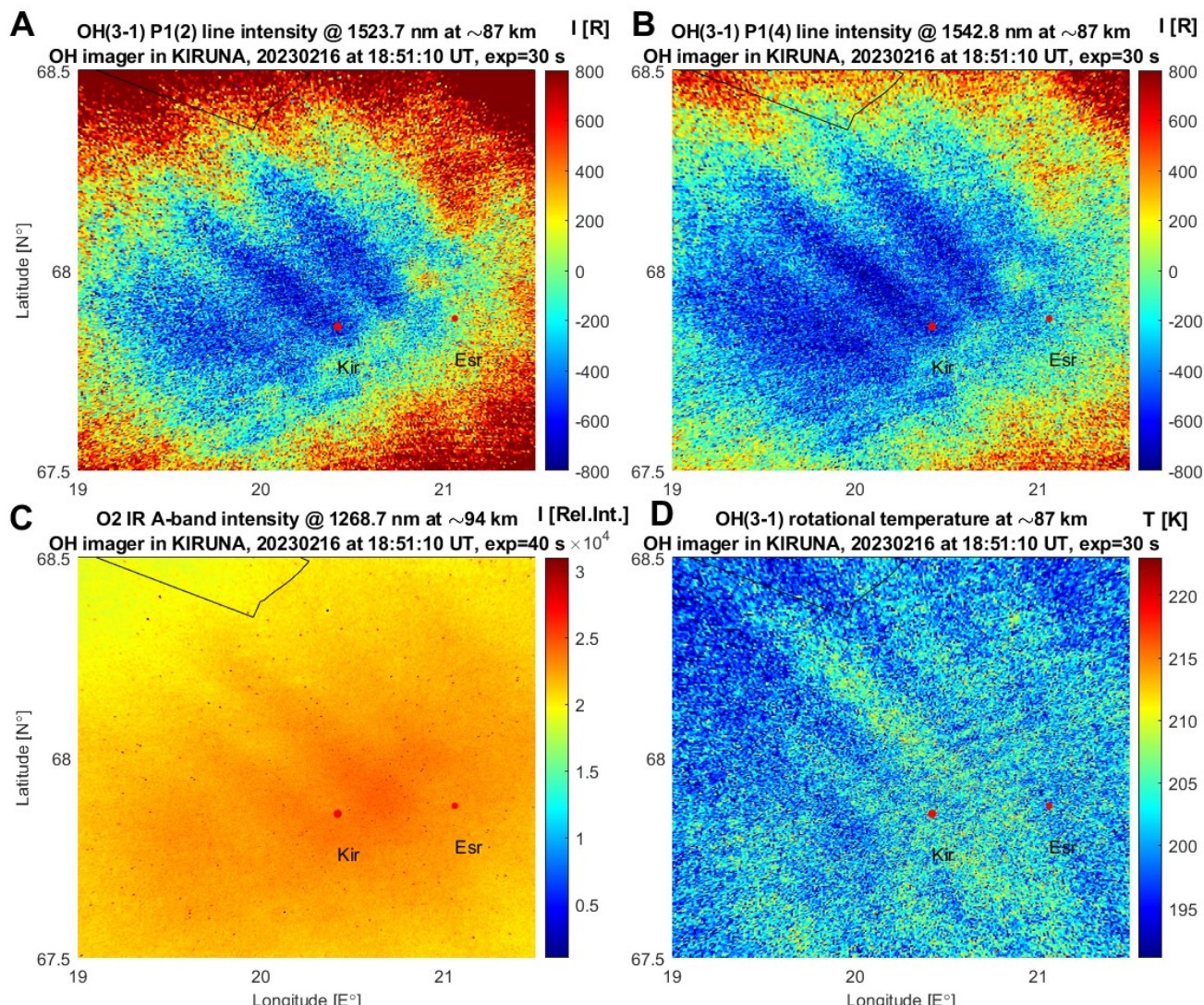

Figure 7: A zoom of the images shown by the black squares in Fig. 6, which focuses on the small-scale wave structure of about 4-6 km wavelength in (A) $P_1(2)$ map, (B) $P_1(4)$ map and (D) temperature map. Residuals of the $P_1(2)$ and $P_1(4)$ intensities are shown after subtraction of the mean intensity values from these zoom images.

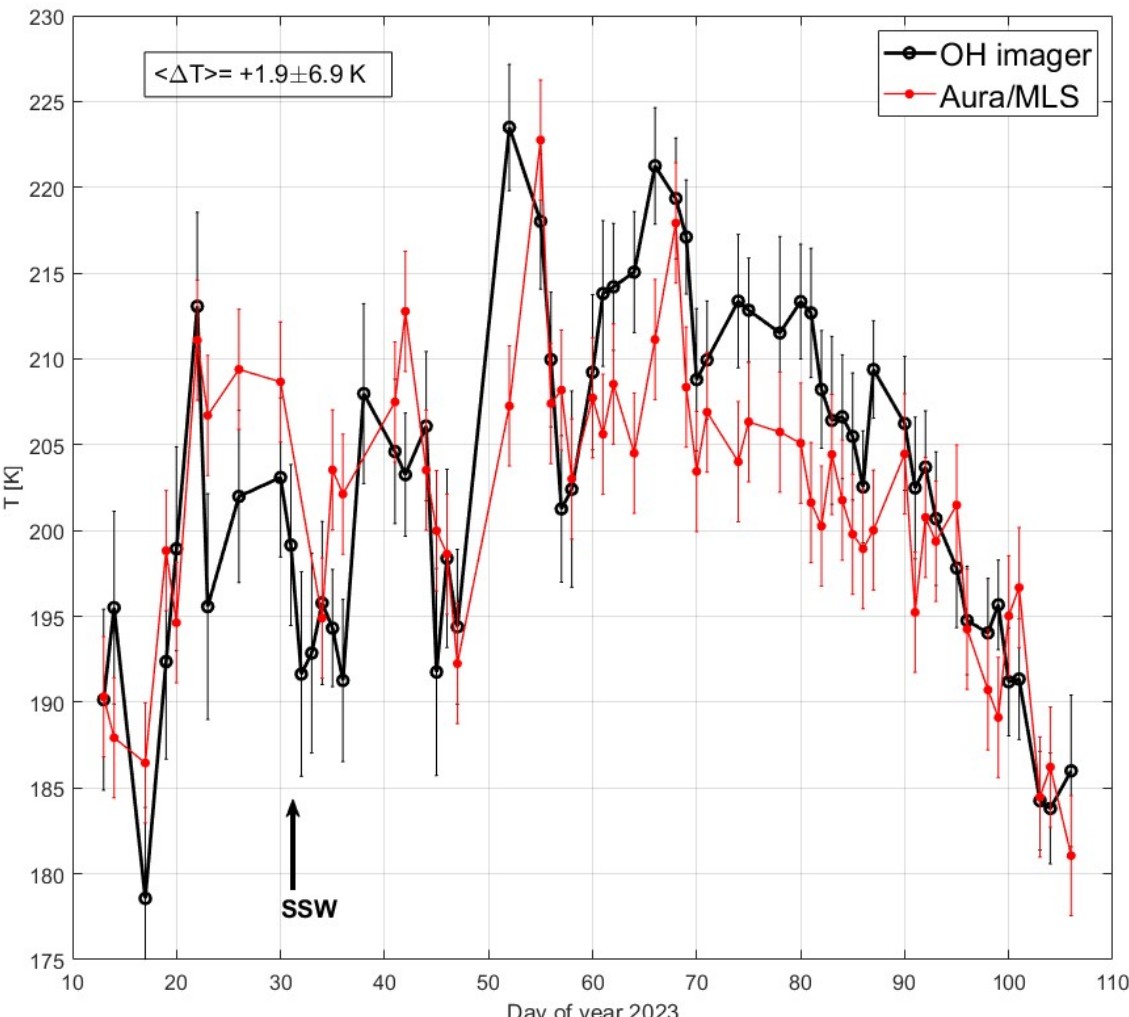

Figure 8: Daily mean OH (3-1) rotational temperature above Kiruna, at zenith, at ~87 km for the 2023 winter season (black circles). Aura/MLS temperature instantaneous measurements close to the Kiruna location at ~86 km (red points). Error bars of the OH (3-1) temperature are one standard deviation for a particular night. Error bars of the Aura/MLS temperature are equal to 3.5 K (see the text). The onset of the sudden stratospheric warming is marked by the black arrow. The considered period is from 13 January until 16 April 2023.