# Peer review of "A novel infrared imager for studies of hydroxyl and oxygen nightglow emissions in the mesopause above northern Scandinavia"

_Atmospheric Measurement Techniques, 2023_

## Referee Comment (RC1)

Review to the manuscript

"A novel infrared imager for studies of hydroxyl and oxygen nightglow emissions in the mesopause above northern Scandinavia"

by P. Dalin et al.

The authors present a new imaging instrument that measures emissions from hydroxyl and molecular oxygen layers in the mesopause region. They describe the technical characteristics of the instrument as well as the derivation of temperatures from these measurements. The new instrument combines the measurements of IR emissions from two different molecules, because the centers of the two emission layers are located at slightly different altitudes. This allows tracing disturbances in the vertical direction additional to the horizontal domain which is enabled due to the imager technique.

Furthermore, they present the first measurements during the winter 2022/23. These measurements were compared with lidar and satellite observations in order to validate the temperatures derived from the imager measurements. Finally, the authors present some small case studies to illustrate the capability of the instrument to monitor temperature changes with time and to detect wave disturbances in both directions vertical and horizontal.

Generally, the manuscript is well structured and written and it addresses scientific questions within the scope of AMT. Thus, I recommend its publication after some minor issues are addressed.

**1 General comments:**

1. The imager shall be able to trace wave disturbances in the vertical domain. As the two emission layers (OH and $O_2$) are not located at a constant altitude and the center altitudes vary with time and season, the distance between the two layers is also not constant. Thus, it should be difficult to obtain absolute information on the vertical propagation. Can you comment and discuss this possible limitation of the technique in some more detail.

2. Typically, the contamination of the lines, especially the P1(4) line, by other emissions such as emissions by the OH(4-2) R-branch is corrected during the temperature estimation process (e.g. Schmidt et al., 2013; Pautet et al., 2014). How is this contamination corrected for your measurements?

3. Could you please clarify the name assignment of the different measurements, because it is a little bit confusing to me. In section 3.4 $I_{P12}$ is introduced as the intensity of the P1(2) line, later it is called the raw intensity and and in Fig. 4 this raw intensity is the sum of P1(2) and BG (background). Is $I_{P12}$ the intensity observed in the spectral range of the the P1(2) line and P1(2) (in Fig. 4 and below Eq. 1) is the real line intensity of the line in counts after subtracting the background? And is the dark noise already subtracted from the measurements?
   Maybe it is helpful to revise these names in the manuscript to get a clear and consistent name assignment.

**2 Specific comments:**

1. Eq. 2: Intuitively, I would expect that the background is subtracted from the measurements in each of the spectral ranges of the observed lines separately. Before this subtraction both single measurements (background and intensity in the spectral range of the emission line) should be corrected for the dark noise influence. Here the dark noise is added (with some factor). Can you please explain a little bit more where the equation comes from.

2. Eq. 3 and Eq. 4: Do the different coefficients $k_i$ have their own uncertainties which then should be taken into account during the error estimation or are the uncertainties too small to have an impact on the total error?

3. Fig. 4: Maybe it is useful to also show the OH equivalent temperature which has been calculated from the lidar observation by vertical averaging in the figure.

4. Fig. 6: It could be helpful to change the ranges of the colour bars as in most cases the full range is not present in the observations and some colours are not used then. This would maybe increase the contrast and visibility of the disturbances.

---

## Author Comment (AC1)

**We thank Reviewer 1 for very useful comments and suggestions which have significantly improved our manuscript. Our detailed replies are mark in bold below. Changes in the revised manuscript are highlighted in yellow.**

Review to the manuscript
"A novel infrared imager for studies of hydroxyl and oxygen nightglow emissions in the mesopause above northern Scandinavia"
by P. Dalin et al.

The authors present a new imaging instrument that measures emissions from hydroxyl and molecular oxygen layers in the mesopause region. They describe the technical characteristics of the instrument as well as the derivation of temperatures from these measurements. The new instrument combines the measurements of IR emissions from two different molecules, because the centers of the two emission layers are located at slightly different altitudes. This allows tracing disturbances in the vertical direction additional to the horizontal domain which is enabled due to the imager technique.

Furthermore, they present the first measurements during the winter 2022/23. These measurements were compared with lidar and satellite observations in order to validate the temperatures derived from the imager measurements. Finally, the authors present some small case studies to illustrate the capability of the instrument to monitor temperature changes with time and to detect wave disturbances in both directions vertical and horizontal.
Generally, the manuscript is well structured and written and it addresses scientific questions within the scope of AMT. Thus, I recommend its publication after some minor issues are addressed.

1 General comments:
1. The imager shall be able to trace wave disturbances in the vertical domain. As the two emission layers (OH and O2) are not located at a constant altitude and the center altitudes vary with time and season, the distance between the two layers is also not constant. Thus, it should be difficult to obtain absolute information on the vertical propagation. Can you comment and discuss this possible limitation of the technique in some more detail.

**These two emission layers (OH and O2) are indeed varied in space and time, making different height distances between these layers. At the same time, if the same wave package, having the same horizontal wavelength and observed phase velocity as well as propagation direction, is observed both in the OH and O2 layers, one can assume that the same gravity wave was propagating both in horizontal and vertical domains. According to the general theory of gravity waves (e.g., Gossard and Hook, 1975) a gravity wave propagates at some angle to the vertical, with tilted phase lines. This should result in an observed phase shift of the same gravity wave between the OH and O2 layers. Once a phase shift and horizontal wavelength are estimated from the OH and O2 maps, one can calculate the vertical wavelength by using the following relation:**

$$\lambda_z = \lambda_x / \tan(\alpha)$$

**where $\lambda_z$ and $\lambda_x$ are the vertical and horizontal wavelengths, $\alpha$ is the angle between wave phase lines and the vertical. Furthermore, if the buoyancy frequency $N$ is a known quantity or is estimated by using lidar or satellite temperature profiles, one can deduce the intrinsic frequency $\omega$ of a gravity wave from the following relation:**

$$\omega = \pm N \cdot \cos(\alpha)$$

Substituting known values of $\omega$, $N$ and $\lambda_x$ into the dispersion relation for gravity waves one can estimate a vertical wavelength again, thus verifying the first estimation of a vertical wavelength. This method is valid for a limited number of gravity waves having vertical wavelengths less that the height distance between the two layers (about 7 km).

Another simple method of the estimation of a vertical wavelength of a gravity wave is based on the assumption that the height difference $D$ between the two layers is a known quantity (Fagundes et al., 1995; Schmidt et al., 2018). If a horizontal phase shift $\Delta\varphi$ of a considered wave package between the both layers is estimated then one can calculate the vertical wavelength $\lambda_z$ using the following relation:

$$\lambda_z = D \cdot 2\pi / \Delta\varphi$$

We will use the both methods to estimate vertical wavelengths of gravity waves propagating through the OH and O2 layers.

We have provided this information in the Discussion of the revised manuscript (lines 444-471).

2. Typically, the contamination of the lines, especially the P1(4) line, by other emissions such as emissions by the OH(4-2) R-branch is corrected during the temperature estimation process (e.g. Schmidt et al., 2013; Pautet et al., 2014). How is this contamination corrected for your measurements?

This is a complicated comment. At present, we do not correct the P1(4) line emission due to emissions by the OH(4-2) R-branch. We should note the following.

We could not find any information of the temperature correction due to emissions by the OH(4-2) R-branch in Pautet et al. (2014). Schmidt et al. (2013) just refer to the method proposed by Lange (1982) when discussing the temperature correction. Schmidt (2016), the doctoral thesis, compares Lange's correction with his own temperature corrections shown in Fig.2.15a in the doctoral thesis. The exact procedure of calculating these temperature correction factors has not been published. A discussion with Dr. Carsten Schmidt (personal communication) suggests that Lange (1982) did not provide details concerning the calculation of the temperature correction and this information is now lost. Temperature corrections by Lange's function as well as presented in the doctoral thesis by Schmidt (2016) were made for the R1(6) line using the value at the line center only. The OH imager registers an integrated intensity of the P1(4) line over a broader range with the interference filter having the spectral width (FMHW) of 2.1 nm. It means that the relative contribution by the R1(6) center line is higher compared to the total area of the R1(6) and P1(4) lines. Summarizing, Dr. Carsten Schmidt suggests that the contribution of R1(6) to the total area of the P1(4) line is less important and the temperature correction might be unnecessary.

Thus, at present, we could not find an explicit published procedure to correct our temperature estimations by the OH(4-2) R-branch for the case of the intensity integrated over the entire spectral width of the P1(4) line.

Also, we should note that a small temperature correction is not important for data analysis dealing with studies of wave disturbances and temperature seasonal changes. A small temperature correction might be important for the temperature validation with other instruments. But in this case, other instruments should also include the same temperature correction that is not obvious.

**Finally, we should note that OH(3-1) rotational temperature estimations agree well with those by the Esrange lidar and Aura/MLS measurements as presented in the manuscript. We might correct the OH(3-1) rotational temperature in the future if we find a correction method appropriate for the OH imager.**

3. Could you please clarify the name assignment of the different measurements, because it is a little bit confusing to me. In section 3.4 $I_{P12}$ is introduced as the intensity of the P1(2) line, later it is called the raw intensity and in Fig. 4 this raw intensity is the sum of P1(2) and BG (background). Is $I_{P12}$ the intensity observed in the spectral range of the the P1(2) line and P1(2) (in Fig. 4 and below Eq. 1) is the real line intensity of the line in counts after subtracting the background? And is the dark noise already subtracted from the measurements?
Maybe it is helpful to revise these names in the manuscript to get a clear and consistent name assignment.

**We have revised the name $I_{P12}$ and $I_{P14}$ lines as being raw intensities, meaning that these raw intensities are without any correction, that is, no subtractions of the atmospheric background and noise have been made. We have provided this information in the revised manuscript (lines 238-240) as well as in the capture to Fig.2 (line 735).**

2 Specific comments:
1. Eq. 2: Intuitively, I would expect that the background is subtracted from the measurements in each of the spectral ranges of the observed lines separately. Before this subtraction both single measurements (background and intensity in the spectral range of the emission line) should be corrected for the dark noise influence. Here the dark noise is added (with some factor). Can you please explain a little bit more where the equation comes from.

**Equation 2 (new Equation 3 in the revised manuscript) comes from Equation 1 in the following way. The $R$ brightness ratio B(P₁(2))/B(P₁(4)) in Equation 1 is in the absolute units (Rayleigh). The instrument registers emission intensities (P1(2), P1(4) and atmospheric background) in relative digital units (counts). In order to relate relative to absolute units, the absolute calibration is performed. The main part of this procedure is to determine filter absolute sensitivities which are different for each filter. These are the coefficients $k2$, $k3$ and $k6$ in Eq.3. In addition, the dark noise is subtracted both from the P1(2) and P1(4) lines as well as from the atmospheric background line which, in turn, is finally subtracted from the P1(2) and P1(4) lines. Since the coefficients $k_2$, $k_3$ and $k_6$ are different this procedure results in different constants (0.22 and 0.60) for the subtracted dark noise in the numerator and denominator in Equation 3. These constants have positive signs since the dark noise is subtracted from the P₁(2) and P₁(4) lines as well as from the atmospheric background line having different coefficients. The coefficients $k_4$ and $k_7$ describe the flat field correction factors being different for each OH (3-1) emission line. Finally, the coefficients $k_1$ and $k_5$ convert photometric units to the Rayleigh, which includes several multiplies and the geometric etendue.
We have provided this information is the revised manuscript (lines 246-258).**

2. Eq. 3 and Eq. 4: Do the different coefficients $k_i$ have their own uncertainties which then should be taken into account during the error estimation or are the uncertainties too small to have an impact on the total error?

**Reviewer 1 is right, the different coefficients $k_i$ have their own uncertainties which should be taken into account during the error analysis. We have provided new error estimations in the revised manuscript (lines 286-299 and new Equation 5).**

3. Fig. 4: Maybe it is useful to also show the OH equivalent temperature which has been calculated from the lidar observation by vertical averaging in the figure.

**We have added the average height-weighted lidar temperature (191.8±12.3 K) as was calculated by vertical averaging across the OH layer, shown by the green asterisk in Fig. 4 (lines 747-749 in the revised manuscript).**

4. Fig. 6: It could be helpful to change the ranges of the colour bars as in most cases the full range is not present in the observations and some colours are not used then. This would maybe increase the contrast and visibility of the disturbances.

**We have changed the ranges of the colour bars in Fig.6 in the revised manuscript.**

---

## Author Comment (AC2)

**We thank Reviewer 2 for very useful comments, suggestions and corrections which have significantly improved our manuscript. Our detailed replies are mark in bold below. Changes in the revised manuscript are highlighted in yellow.**

Review of "A novel infrared imager for studies of hydroxyl and oxygen nightglow emissions in the mesopause above northern Scandinavia - Peter Dalin et al. 2023"

The authors present a new imaging system that measures the OH and O2 airglow emission originating from two different altitude layers in the middle atmosphere. The imager is equipped with overall five filters, four interference filters and one dark filter. Two of the interference filters are for the observation of OH(3-1) P1(2) and P1(4) to derive the rotational temperature in about 87km altitude, one for to observation of the O2 emission layer in about 94km, and one for the background. An additional dark filter is for corrections of the dark current. One complete cycle through all filters is done every 2.75 minutes. The instrument is thoroughly calibrated with an absolute calibration considering the imager and all the optical components. A geometrical calibration is used for georeferencing the all-sky observations.

The OH rotional temperature is derived from the two observed OH(3-1) emission lines following a procedure based to the one from Pautet et al. (2014), but slightly modified with new Einstein A-coefficients. An error analysis for the temperature derivation is done and shown in detail. The derived temperatures are compared with the temperature observations of a nearby LIDAR instrument at Esrange in about 40km distance of the new imager's site for the imager's pixel location nearest to the LIDAR and the altitude of the OH layer in the LIDAR data. The author's find a good agreement of both measurements within about 3 K for a particular night and only -0.2K±1.6K on average. Additionally, they compare the temperatures derived from the imager with Aura/MLS satellite measurements and find an average temperature difference of 2.8K±7.8K. Considering the different field-of-views and spatial distances between ground-based instrument and satellite this seems reasonable.

In a case study the authors highlight the observation of a ripple structure visible in the OH temperature map with about 4-6 K temperature amplitude and with a horizontal wavelength of 4-8 km. This shows nicely the potential of the imager to resolve small-scale structures in the temperature maps. Also, they show a medium-scale gravity wave which is visible in the OH and O2 layer simultaneously. Analyzing daily mean OH rotational temperatures from January to April 2023 the authors see seasonal temperature variations typical for the site's location. They also see peaks in the data and speculate whether these could be due to the sudden stratospheric warming event starting in February 2023.

Overall the author's present a valuable new imager instrument for atmospheric observations in the middle atmosphere utilizing two airglow emissions. Especially the combination of OH temperature maps and observing the O2 layer quasi simultaneously offers great opportunities for future investigations of gravity waves and many other phenomena. The manuscript is well written and the analyses described concisely. I recommend it to be published in AMT after addressing the following few minor points.

General remarks:

1. I suggest a short paragraph about currently used imagers to emphasize the novel aspects of the presented imager instrument. E.g. many imagers operate in the VIS-NIR range (e..g Li et al. 2005), but also many in the SWIR range. When integrating over larger

parts of the SWIR range the integration time is often of the order of a second (Hannawald et al. (2016)). There are also other instruments utilizing a filter wheel to observe different airglow emissions simultaneously, but often integrating over larger parts of e.g. the OH bands (e.g. Mukherjee et al. 2010), so that no temperature derivation is possible. On the other hand, Pautet et al. (2014) uses the AMTM to derive temperatures (which was shortly mentioned in the manuscript). Other imager types focus on very small scales (e.g. Sedlak et al. (2016), Hecht et al. 2023 (https://doi.org/10.1029/2023JD038754)). However, none of these instruments - as far as I am aware of - derives temperature and uses multiple emission layers. It would be valuable to work out the differences of the new imager to current imagers.

**More than fifty sites conducting spectroscopic and imaging airglow observations are presented at the Network for the Detection of Mesospheric Change (NDMC) which is a global program investigating climate change signals in the mesopause region (https://ndmc.dlr.de/). Li et al. (2018) have summarized a global distribution of all-sky airglow imager sites (see Table 1 in their paper and references therein). Some of OH spectrographs and imaging instruments have a narrow field of view of 30 degrees and less such as the Ground-based Infrared P-branch Spectrometer instrument (GRIPS 6) in Oberpfaffenhofen, Germany (Schmidt et al., 2013), the Aerospace Nightglow Imager 2 (ANI2) in the Andes, Chile (Hecht et al., 2023), and the Spectral Airglow Temperature Imager (SATI) in Resolute Bay, Canada (Wiens et al., 1997). A number of OH imaging instruments measure OH emissions in relative units without mesopause temperature derivations such as the ANI2, OH all-sky airglow imager in Kazan, Russia (Li et al., 2018), and the Fast Airglow IMager (FAIM) in Oberpfaffenhofen, Germany (Hannawald et al., 2016). Some of the OH imagers register OH emissions (including temperature derivations) without capturing infrared emissions coming from the $O_2$ layer such as the Advanced Mesospheric Temperature Mapper (AMTM) at South Pole (Pautet et al., 2014), and the Near InfraRed Aurora Camera (NIRAC) in Svalbard, Norway (Nishiyama et al., 2024). In the present paper, we describe a novel infrared wide-angle imaging instrument capable of registering emissions coming from two emitted layers (OH and $O_2$) and deriving the mesopause temperature. The OH imager is the first one of its kind installed in northern Scandinavia.**

**We have provided this information in the revised manuscript (lines 71-89).**

2. The shown images and the temperature map look very nice and highlight the capabilities of the instrument. However, I imagine that one or a few video sequences of the shown night as supplement data might by quite more impressive and useful to show the potential of the instrument.

   **We agree with it and have added a video sequence (OH_imager_video_160223.avi) of the shown night as supplement data which can be obtained from the Swedish National Data Center (https://snd.gu.se/en/catalogue/dataset/preview/d5776ec9-d346-4e6a-a8c3-1445e75201b6/1**

**Specific remarks:**

L44-45: "hot topic of current atmospheric research." It would be nice to have more up to date articles . Wüst et al. (2023, https://doi.org/10.5194/acp-23-1599-2023) might be a good fit.

**We have updated "hot topics" up to date in the revised manuscript (line 44).**

L136: The re-imaging lens was not described in the setup before while the primary lens and telecentric lens were. Could you also describe it shortly?

**The reimaging optics consist of a doublet field lens, and a combination of a second doublet and a f/1 compound lens in front of the sensor. We have added this information in the revised manuscript (lines 152-154).**

L146-147: "…implying that the instrument can readily resolve the highest frequency range in the gravity wave spectrum." Waves with frequencies near the Brunt-Väisälä-Frequency $(2*pi/(5 * 60) = 0.021$ assuming a BV-period of 5min at the mesopause) might be aliased as the Nyquist frequency of the instrument is $f\_ny = (0.5 * 2*pi / (2.75 * 60) = 0.019$. Especially small-scale gravity waves tend to have periods near the BV-period (see e.g. Tang et al. 2014). I suggest to formulate the sentence a little more cautiously. In this context is it even nicer to see the ripple structure proposed later in the manuscript to show the potential of the instrument even for small-scale variations.

**Reviewer 2 is right, a BV-period might be aliased with the Nyquist frequency of the instrument. But we do not intent to study such extremely high frequencies which are very close to the Brunt-Väisälä frequency. In general, the spectral analysis is applied to frequencies which are less than the Nyquist frequency. Besides, a BV-period is varied in the mesopause around 5 minutes. That is why our intension is to study gravity waves having observed periods around 10 minutes and more. We have added this information in the revised manuscript (lines 163-167).**

L184-185: How was the flat-field correction done? Was a black-body source used or an average of measurements or something different? I can imagine it wasn't easy to perform a flat-field correction with a 120° field-of-view.

**The flat-field is a complicated procedure but basically it consists of three steps:**
1. **The cylindrically symmetric component of the lenses is characterized first. This is done on an optical bench (180° rotary table) and an integrating sphere, where the sphere is illuminated with a tunable laser at the emission wavelengths of each of the two OH channels. This is done with an uncoated glass blank in the optical path to have the exact same optical system, but without the filter contribution.**
2. **Then the imager is inserted into the integrating sphere. An additional diffuser is placed over the fisheye lens. Images are taken with the filters in place and with the uncoated glass blank in place. This is to find the attenuation of the filters relative to the uncoated glass blanks.**
3. **Zernike polynomials are then used for the fitting to produce the smooth flat fields. We have provided a short description of this procedure in the revised manuscript (lines 206-213).**

L193-197, L204: Can you give some additional information about the geometrical calibration? Unfortunately, in Dalin et al. (2015) I couldn't find more details on the geometrical calibration with reference stars (also not in the publication Dubietis et al. (2011) therein) which is directly related to the geometrical calibration done here. E.g. have you used an infrared star catalogue to find reference stars as you observe in the SWIR range rather then the VIS-NIR? What form has the 3rd order polynomial?

**The PPM Star Catalogue (Positions and Proper Motions Star Catalogue) was used to identify positions of the reference stars. It contains positions and proper motions of 378,910 stars on the whole sky in the J2000 coordinate system. Since the majority of stars emit light in the VIS-NIR as well as in SWIR range, it was possible to identify 50 reference stars on images with the wide-band filter (1000-1600 nm).**

**The 3$^{rd}$ order polynomial has the following form:**

**$P = a_1 \cdot x^3 + a_2 \cdot x^2 y + a_3 \cdot y^2 x + a_4 \cdot y^3 + a_5 \cdot x^2 + a_6 \cdot xy + a_7 \cdot y^2 + a_8 \cdot x + a_9 \cdot y + a_{10}$**

**where ten $a_i$ are free coefficients determined under solving this equation in the least-squared sense, x and y are horizontal coordinates of the reference stars on the analyzed image. We have provided this information in the revised manuscript (lines 220-225).**

L209-210: For me it is not clear where this formula is coming from, especially the meaning of k_1 to k_7. Is k_4 e.g. the standard deviation of the noise n_dc? Why is there a different parameter for n_dc in denominator and numerator? What is meaning of the scaling factors k_1 and k_5? I think that formula would be much clearer with a few lines about what the coefficients mean and how they are "determined in a laboratory".

**Equation 2 (new Equation 3 in the revised manuscript) comes from Equation 1 in the following way. The *R* brightness ratio B(P$_1$(2))/B(P$_1$(4)) in Equation 1 is in the absolute units (Rayleigh). The instrument registers emission intensities (P1(2), P1(4) and atmospheric background) in relative digital units (counts). In order to relate relative to absolute units, the absolute calibration is performed. The main part of this procedure is to determine filter absolute sensitivities which are different for each filter. These are the coefficients *k2*, *k3* and *k6* in Eq.3. In addition, the dark noise is subtracted both from the P1(2) and P1(4) lines as well as from the atmospheric background line which, in turn, is finally subtracted from the P1(2) and P1(4) lines. Since the coefficients $k_2$, $k_3$ and $k_6$ are different this procedure results in different constants (0.22 and 0.60) for the subtracted dark noise in the numerator and denominator in Equation 3. These constants have positive signs since the dark noise is subtracted from the P$_1$(2) and P$_1$(4) lines as well as from the atmospheric background line having different coefficients. The coefficients $k_4$ and $k_7$ describe the flat field correction factors being different for each OH (3-1) emission line. Finally, the coefficients $k_1$ and $k_5$ convert photometric units to the Rayleigh, which includes several multiplies and the geometric etendue.**
**We have provided this information is the revised manuscript (lines 245-258).**

L222: I may be wrong, but I would calculate the angular side length of a single pixel with 120° / 512 pixels = 0.23 °/pixel.

**Here we present a single-pixel area (0.003 x 0.003°) as projected onto the Earth's surface. This is not the angular side length of a single pixel.**

L333: "due to small-scale gravity waves (ripples)…". Following Li et al. (2017) and references therein, ripples are not gravity waves rather than instability structures and are often referred to as "wavelike structures" or simple "ripples". It might be difficult to distinguish between small-scale gravity waves and ripples. I suggest rephrasing to make clear that these are different phenomena.

**In general, both small-scale gravity waves and ripples are atmospheric gravity waves since a restoring force is the buoyancy force acting on any disturbed air parcel in small-scale waves and ripples, at least at the beginning phase of their formation. But due to multiple definitions of ripples we have removed "ripples" from the revised manuscript, which is not important for this content.**

L476: The doi of Dalin eta al 2015 seems to be wrong, I think it should be https://doi.org/10.1002/2014GL062776

**Reviewer 2 is right, the doi number has been corrected.**

Figure 6: I wonder about the same timestamp in the title of all 3 emission plots (6a-c). I would expect 30-32s gaps between P1(2), P1(4) and O2.

**Reviewer 2 is right, there are 30 s gaps between P1(2), P1(4) and O2 maps in Fig.6. The presented timestamp corresponds to the time of the middle point of a current filter wheel cycle which takes 2.75 min. We consider it is better to present the same timestamp for all maps shown in Fig.6 for consistency purposes as well as it is very convenient for a successive time series analysis. We have added this information to the capture of Fig.6 (lines 769-771).**

Additionally, I can't recognize the small wave structure of 4-8 km in P1(2) or P1(4) which could be an issue of contrast/color range (I can see it in the rotational temperature map). I suggest to add another figure with a small crop/zoom of the whole images focusing on the small-scale structure and adjusted color range.

**We have added another figure (new Figure 7) with a zoom of the whole image focusing on small-scale structures of 4-6 km wavelength and have adjusted color ranges in Figs. 6 and 7.**